# Characterization of AZ31/HA Biodegradable Metal Matrix Composites Manufactured by Rapid Microwave Sintering

**DOI:** 10.3390/ma16051905

**Published:** 2023-02-25

**Authors:** Shivani Gupta, Apurbba Kumar Sharma, Dinesh Agrawal, Michael T. Lanagan, Elzbieta Sikora, Inderdeep Singh

**Affiliations:** 1Mechanical and Industrial Engineering, Indian Institute of Technology Roorkee, Roorkee 247667, India; 2Materials Research Institute, Pennsylvania State University, State College, PA 16802, USA

**Keywords:** microwave sintering, biodegradable composites, mechanical properties, corrosion resistance, biodegradability

## Abstract

This study reports the development of magnesium alloy/hydroxyapatite-based biodegradable metal matrix composites (BMMCs) through rapid microwave sintering. Magnesium alloy (AZ31) and hydroxyapatite powder were used in four compositions 0, 10, 15 and 20% by weight. Developed BMMCs were characterized to evaluate physical, microstructural, mechanical and biodegradation characteristics. XRD results show Mg and HA as major phases and MgO as a minor phase. SEM results correlate with the XRD findings by identifying the presence of Mg, HA and MgO. The addition of HA powder particles reduced density and increased the microhardness of BMMCs. The compressive strength and Young’s modulus increased with increasing HA up to 15 wt.%. AZ31-15HA exhibited the highest corrosion resistance and lowest relative weight loss in the immersion test for 24 h and weight gain after 72 and 168 h due to the deposition of Mg(OH)_2_ and Ca(OH)_2_ layers at the sample surface. XRD analysis of the AZ31-15HA sintered sample after an immersion test was carried out and these results revealed the presence of new phases Mg(OH)_2_ and Ca(OH)_2_ that could be the reason for enhancing the corrosion resistance. SEM elemental mapping result also confirmed the formation of Mg(OH)_2_ and Ca(OH)_2_ at the sample surface, which acted as protective layers and prevented the sample from further corrosion. It showed that the elements were uniformly distributed over the sample surface. In addition, these microwave-sintered BMMCs showed similar properties to the human cortical bone and help bone growth by depositing apatite layers at the surface of the sample. Furthermore, this apatite layer can enhance osteoblast formation due to the porous structure type, which was observed in the BMMCs. Therefore, it is indicative that developed BMMCs can be an artificial biodegradable composite for orthopedic applications.

## 1. Introduction

Demand for magnesium alloys and their composites is increasing in medical science due to their lightweight, osteoconductive, nontoxic, and biodegradable attributes. However, the high corrosion rate of pure magnesium in physiological conditions makes it inappropriate for permanent orthopedic implants. On the other hand, magnesium-based artificial materials are the most suitable for biodegradable orthopedic fixation aids, such as screws, bone plates, stents, rods, and scaffolds, in which degradability is the most desired attribute. Biodegradable materials assist tissue growth surrounding the broken joint and get degraded after therapeutic action. Therefore, such materials reduce the risk of secondary trauma and stress shielding effects that are generally witnessed in the case of non-degradable biomaterials. Consequently, surgeons have started to use biodegradable materials in orthopedic implantations to minimize patient pain and inconvenience.

However, limited metals, ceramics, and polymers with biodegradable characteristics are available and suitable for implantation in the human body. Magnesium and its alloys are widely used as biodegradable metals for medical applications since it is an essential mineral in the human body and has excellent biodegradability [1]. The total magnesium content in a healthy human body is approximately 25 g, most of which is in bone and soft tissues. Magnesium deficiency can cause insulin resistance, metabolic syndrome, heart disease, and osteoporosis; indirectly, magnesium deficiency can reduce calcium and potassium levels in the blood [2,3,4,5]. Similarly, hydroxyapatite (Ca_10_(PO_4_)_6_(OH)_2_) is a bio-ceramic characteristically similar to human bone. Therefore, Mg and HA composites were explored as potential biodegradable materials for orthopedic implants [6]. Due to its potential, there has been significant growth in syntheses, economic processing of these materials, and understanding of characteristics [7]. However, the fabrication of composites with the necessary characteristics is challenging.

It was reported that pure Mg and its alloys could not be directly applicable in the human body environment due to their high degradation rate. Therefore, studies were carried out on improving the corrosion resistance of Mg-based biomaterials coated with HA, Ti alloys, and ZrO_2_; better corrosion resistance and enhanced biocompatibility were reported [8,9,10]. Magnesium-based biocomposites with HA, PLA, bioglass, etc., were also found as potential materials for orthopedic fixation aids. Therefore, many studies have focused on processing these composites using powder metallurgy, casting, extrusion, and friction stir processing [11,12,13,14,15,16,17]. Composites like Mg/HA, Mg-Zn-Zr/HA, AZ91/HA, and ZK60A/CPP were also fabricated with improved hardness, compressive strength, and bio-corrosion resistance with in vitro biocompatibility [18,19,20,21,22,23,24,25]. Jaiswal et al. fabricated Mg-3Zn-HA BMMC using conventional sintering (CS) with different HA contents and reported that 5 wt.% HA composition showed the highest corrosion resistance in simulated body fluid (SBF) and adequate compressive strength along with improved osteogenic cell adhesion properties [26]. Later, Mg/HA BMMCs were developed from a high-frequency induction sintering technique and an increase in relative density, compressive strength, microhardness, and crystal size was observed when the sintering temperature was increased [27]. Mg/HA BMMCs of 10, 20, and 30 wt.% of HA were also developed using hot pressing and extrusion. It was reported that 10 wt.% HA sample exhibited higher strength. In contrast, the 30 wt.% HA sample obtained higher resistance in corrosion than 10 wt.% HA with better cytocompatibility [28]. In addition, magnesium alloy AZ91/fluorapatite composites were also developed using powder metallurgy with improved corrosion resistance, mechanical properties, and osteoconductivity [29]. In addition, AZ31/nano HA composites were fabricated from friction stir processing (FSP) and reported that fine grain structure was obtained with the integration of nano HA. Reinforcement of nano HA particles enhanced the cell corrosion resistance and adhesion properties of composites in vitro tests [30]. Yee-Hsien Ho et al. fabricated AZ31/HA composites using a friction stir additive manufacturing technique and observed that the addition of HA particles refined grain structure and improved biomineralization during the immersion test. It also reported good biocompatibility in higher HA content composition [31]. Other than FSP, conventional powder metallurgy processes like spark plasma sintering (SPS) are widely explored in composites processing. The microwave sintering (MS) technique is used in limited studies to process pure HA bio-ceramic in comparative studies with conventional sintering. It is found that microwave-sintered composites exhibited higher compressive strength, reduced porosity, and smaller grain sizes (33–50%) than conventionally sintered composites [32,33].

Therefore, there is ample scope to explore microwave-assisted powder metallurgy routes in the fabrication of magnesium alloy-based BMMCs. This study used magnesium alloy (AZ31) and hydroxyapatite for metal matrix composites since this alloy has exhibited a nontoxic nature due to its low aluminum presence [34]. Further, microwave-sintered AZ31-HA BMMCs were characterized for density, phase analysis, microstructural evaluation, mechanical properties, and in vitro biodegradable behavior.

## 2. Materials and Experimental Procedure

The details of the materials used, the experimental procedure followed for developing the microwave-sintered magnesium alloy/HA-based metal matrix biodegradable composites, and their characterizations are presented in the following sections.

### 2.1. Material Details

AZ31 powder (Nextgen Steel & Alloys, Mumbai, India) of average particle size of 40 µm, and hydroxyapatite (Aldrich Chemical Company Inc., Milwaukee, WI, USA) of agglomerated particles of submicron size of 44 µm were used as precursors. Table 1 lists the chemical composition of AZ31 alloy. Typical XRD spectra of AZ31 and HA powders are shown in Figure 1. The XRD results individually confirmed the presence of α-Mg phase in AZ31 and Ca_10_(PO_4_)_6_(OH)_2_ phase in HA powder of JSP. Earlier studies revealed good results from 10 to 20 wt.% of HA [35,36]. Therefore, this study was focused on four different compositions, such as 0, 10, 15 and 20 weight percent of HA with AZ31. Further, the powders were mixed in polyethylene bottles using MgZr cylindrical balls, rotating at 100 rpm for 12 h. The powder mixtures were compacted into green pellets (ϕ13 mm × 3 mm) using a cold static hydraulic pressure of 450 MPa.

### 2.2. Development of AZ31/HA Composites

The compacted green pellets of different compositions were sintered in a multimode microwave tube furnace operating at 2.45 GHz frequency and 3 kW maximum power. All samples were sintered in a working environment of forming gas. Table 2 represents the processing conditions used during the development of BMMCs.

## 3. Characterization Details

Microwave-sintered BMMCs were characterized by their physical, metallurgical, mechanical and corrosion behavior; details are discussed in the following sections.

### 3.1. Evaluation of Physical and Metallurgical Properties

The density of the developed BMMCs was calculated from the mixing rule followed by Archimedes’ principle. Four readings were taken to calculate the experimental density of the BMMCs. X-ray diffractometry (Make: Malvern Panalytical, Model: Empyrean 3rd generation) was then used to estimate the phases formed in the sintered BMMCs and after corrosion testing. The used 2θ range was 20° to 90° at a scanning speed of 1°/min. The XRD results were confirmed and correlated with the microstructural results. Samples for microstructural analysis were prepared using waterproof sandpapers from 1000 to 2500 mesh size, followed by cloth polishing using colloidal silica of 0.4 µm. The samples were polished on a mechanical polishing machine (Made: Allied High Tech Products Inc.,Rancho Dominguez, CA, USA). Further, scanning electron microscopy (Make: Thermo Scientific, Model: Verios) was used for microstructural analysis at 15 kV and 500X. Additionally, elemental mapping was also extracted from the energy-dispersive X-ray spectroscopy (EDS) analysis that confirmed the presence of existing elements (Mg, Al, Zn, Ca, P, and O) as a result of various major phases such as Mg, MgO and Ca_10_(PO_4_)_6_(OH)_2_.

### 3.2. Evaluation of Mechanical Properties

The present study measured microhardness, compressive strength and Young’s modulus to assess the mechanical properties. The BMMCs samples were prepared using waterproof polishing since magnesium is highly reactive with water. Ten indentations were carried out for each composition using (ASTM standard C 1327-99) Vickers’s microhardness tester (Make: Chennai Metco Pvt. Ltd., Chennai, India). These indentations were performed radially from edge to edge. The average value was considered and reported. Further, a Universal Testing Machine (Model: Instron 5982, Make: Instron Corporation, Norwood, MA, USA) was used to measure the compressive strength and Young’s modulus of the sintered BMMCs; the associated scanning rate was 0.2 mm min^−1^, and ASTM E9 standard was used.

### 3.3. Evaluation of Corrosion Behavior of BMMCs

Assessment of corrosion resistance is essential for any biodegradable material as it can decide the degradation rate of the developed material in actual working conditions. Therefore, the following measurements were carried out on three samples of each composition.

#### 3.3.1. Immersion Test for Measurement of Weight Loss

The immersion test on microwave-sintered BMMCs was carried out to quantify the weight loss at 37 ± 2 °C in Hanks’ Balanced Salt Solution (HBSS) as per the standard ASTM-G31-72. The composition of HBSS ions (in mmol/L) is listed in Table 3. For comparison, the composition of the blood plasma is also included in the table. The ratio of immersion solution volume to sample area used was 20 mL/cm^2^. A sample of each composition was immersed in separate flasks filled with HBSS at 37 ± 2 °C with 7.2 ± 0.2 pH in open-air conditions. The weight loss samples were placed on the marble balls to maximize the exposed area in the HBSS electrolyte. The exposed surface area of the cylindrical sample and relative weight loss was calculated using Equations (1) and (2), respectively.
(1)$$A_{s}=2\pi r^{2}+2\pi rh$$
(2)$$W=\frac{\left(w_{b}-w_{a}\right)}{w_{b}}\times 100_{\;}$$
where *A_s_* is the exposed surface area of the cylindrical sample, *r* is the radius (0.65 cm) and *h* is the thickness (0.2 cm). The calculated exposed surface area is 3.47 cm^2^. *W* is the relative weight loss of immersed sample after 24 h, *w_b_* and *w_a_* are the weights of samples before and after the immersion test. The corrosion rate was calculated from Equation (3).
(3)$$CR=\frac{87.6\times \Delta W}{A_{s}\times \rho \times t}$$
where *CR* is corrosion rate (mm/year), ∆*w* is weight loss (milligrams), *A_s_* is the exposed surface area (cm^2^), *ρ* is the density of sample (g/cm^3^) and *t* is the exposure time (h).

#### 3.3.2. Measurement of pH and Mg^2+^ Ion Concentration in HBSS

The pH value of HBSS bulk solution after 24 h of the immersion test was measured by a pH meter (Make: Thermo Scientific, Waltham, MA, USA), with the pH and temperature measurements ranging from 0 to 14 and 0 °C to 50 °C, respectively. Inductively coupled plasma mass spectroscopy (Make: Thermo Scientific, Model: iCAP TQe Quadrupole, USA) was used to measure Mg^2+^ ions concentration in an immersion solution. After 24 h, the samples of each composition were removed from the HBSS, gently rinsed with DI water, cleaned with chromium trioxide, and then dried to measure the relative weight loss of BMMCs immersed in HBSS.

#### 3.3.3. Electrochemical Analysis

Electrochemical analysis of various BMMCs was carried out using a potentiostat (Made: Gamry Instruments, Philadelphia, PA, USA, Model: Reference 600). Samples of each composition were prepared by welding copper wire to the surface and mounted in epoxy. Samples were then polished using silicon carbide paper of grades 1000 to 2500, and finally polished with colloidal silica of 0.04 µm particle size. The exposed sample surface area was 1.32 cm^2^. The prepared samples were set as a working electrode to measure the corrosion behavior. The graphite was employed as a counter and saturated calomel electrode (SCE) as reference electrodes to complete the 3-electrode electrochemical cell setup. The cell was filled with HBSS solution and coupled with a heating system containing a heating plate, thermocouple and temperature controller that was used to maintain the solution temperature at 37 ± 2 °C. Before electrochemical measurements, the samples were immersed in the HBSS solution for 30 min for open circuit potential stabilization. Potentiodynamic polarization was carried out at a scan rate of 1 mV/s within the potential range of –1.8 V to 0 V vs. SCE. Corrosion current (I_corr_) and corrosion potential (E_corr_) were calculated from the Tafel curve.

#### 3.3.4. Observation of Surface Morphology

Following the corrosion test, a sample of BMMCs was kept out of the solution for phase analysis through XRD. SEM and EDS analyses were carried out to identify the surface morphology and the possible elements at the sample surface after the immersion test.

## 4. Results and Discussion

The AZ31-HA biocomposites (AZ31-0HA, AZ31-10HA, AZ31-15HA, and AZ31-20HA) were successfully developed using microwave sintering at a microwave power of 650 W in forming gas with the direct microwave heating method. Forming gas helped to reduce MgO formation by providing a reduced atmosphere. Hydrogen suppresses the formation of oxides that could form because of the presence of OH^−^ in HA. The time-temperature curve of the AZ31-15HA BMMC sintered at 500 °C with 10 min sintering time is shown in Figure 2. AZ31-15HA composition showed the best properties among the other compositions. AZ31-15HA (Figure 2) revealed the characteristics of different heating zones. The sample temperature was recorded using an infrared pyrometer, which records temperatures above 250 °C. The slope (θ_1_) of the heating curve at point A in zone-I is considerably high. However, in zone II, the heating rate is very slow (θ_2_ << θ_1_ << θ_3_) in comparison to zones I and III. The profile thus indicates that the material increases in temperature during zone-I due to less microwave absorption by the hydroxyapatite particles and more reflection from the metal alloy particles. In zone II, however, the temperature of the material increases very slowly due to less microwave power exposure (min 330 W) and the slope of the heating curve 2.2 was calculated at zone midpoint B. While increasing microwave power slowly (min 350 W to max 650 W), the temperature increases, the sample achieves sufficient temperature for coupling with the microwaves, and rapid heating starts resulting in the observed higher rate of temperature rise (θ_3_ = 25.6). The sintered material gets softened during this period, followed by the onset of necking and bonding in zone IV for 10 min sintering time. In addition, Figure 3 shows the mechanism of microwave sintering and the interaction of microwave radiations and AZ31 metal alloy and HA ceramic powder particles. The HA ceramic particles absorb more microwave energy than the AZ31 alloy. This is attributed to the differences in the dielectric properties of the alloy and the ceramic. The HA has high dielectric loss because of the presence of hydroxyl ions which leads to volumetric heating; while the AZ31 is metal-based and behaves as a reflective material during microwave exposure. In case (a) in Figure 3, the material with less HA content experiences non-uniform heating. The HA particles, located fairly apart, behave like tiny localized heat sources by absorbing microwave energy. Further, these heat sources transfer heat to metal particles and help in raising the temperature of metal alloy particles primarily by a conductive mode of heat transfer. Conversely, the addition of more HA content per unit volume of the material makes the density of the localized heat sources higher resulting in a fairly uniform distribution of the sintering temperature in a relatively short time as observed during sintering and illustrated in Figure 3 (case (b). This results in more uniform necking and sintering. Excessive addition of HA, however, might result in localized hotspots or overheating in the material, causing eventual deterioration of the sintered material as evidenced and discussed in the subsequent sections. The developed BMMCs were characterized to measure physical, mechanical, and metallurgical properties and corrosion behavior.

### 4.1. Density and Metallurgical Properties

Figure 4 shows the relative density of AZ31-HA biocomposites. It is observed that the AZ31-0HA sample exhibits the highest relative density of 99.43% in a sintering time of 10 min, while AZ31-20HA shows the lowest relative density. The reduction in density is due to the integration of HA particles, which hinders the densification of biocomposites [38]. HA particles are harder than AZ31 particles, which are difficult to compress during compaction. In addition, the formation of MgO after sintering may also reduce the relative density of final sintered biocomposites. The difficulty in achieving higher densification of HA at higher temperatures has been reported by other researchers who could not achieve more than 95% relative density [39]. Therefore, the relative density of microwave-sintered AZ31-HA composites depends on HA content. The relative porosity percentage in BMMCs is 0.57%, 1.63%, 2.12% and 5.16% for AZ31-0HA, AZ31-10HA, AZ31-15HA and AZ31-20HA, respectively.

Figure 5 presents the XRD results of microwave-sintered biocomposites. Major diffraction peaks belong to Mg and hydroxyapatite; a minor phase of MgO also exists. Magnesium is highly reactive with atmospheric oxygen and needs extra care to prevent oxidation at room temperature. OH^−^ group present in HA evaporates at around 100 °C and may react with Mg to form MgO [38,39]. Moreover, no other peaks are found that indicate the absence of a chemical reaction between AZ31 and HA at 500 °C sintering temperature. It helps to maintain the bioactivity of AZ31-HA BMMCs.

Figure 6 shows the backscattered electron microscope images of microwave-sintered BMMCs, which also exhibit the same results and correlate with the optical results. These results illustrate that the HA clusters occupied space in the matrix around the metal alloy particles and contributed to improving the properties of the BMMCs. Figure 6a AZ31-0HA sample indicates the presence of magnesium and pores at the surface. In contrast, Figure 6b–d shows SEM micrographs of AZ31-10HA, AZ31-15HA and AZ31-20HA BMMCs samples. These samples were analyzed on the backscattered detector and observed magnesium grains and the distribution of nano-HA particles at the grain boundaries of Mg grains. Black spots were observed in the microstructural images that resemble similar pores. Therefore, these black spots were analyzed at higher magnification (400 nm) and we found that they are not pores. They are agglomerated nano-HA particles that settled in between Mg grains and are distributed in the matrix. Moreover, Al and Ca particles are also observed at the sample surface, which is identified by EDX results shown in Figure 7.

Figure 7 represents the elemental mapping of microwave-sintered AZ31-HA composites with the percentage of elements present in the EDX samples. These results confirm the Mg, O, Ca, P and Al distribution. Zn is present in a very small weight percentage, less than the detectable limit in elemental mapping, while it is found in the elemental percentage data. EDX results depict Mg as green in color and uniformly distributed. It also has proper grain size. Conversely, Ca, P and O are available as HA at the grain boundaries of magnesium due to the nano size of HA particles. Al and Ca are observed as free elements.

### 4.2. Mechanical Properties

Figure 8 shows the microhardness data of AZ31-HA BMMCs with their respective optical images of indentation. The average microhardness values of 49.23 ± 1.95 HV, 57.96 ± 1.56 HV, 61.42 ± 1.65 HV and 65.86 ± 1.89 HV are obtained for AZ31-0HA, AZ31-10HA, AZ31-15HA and AZ31-20HA, respectively. The data reveal that the microhardness of BMMCs increases with increased HA content. HA is a ceramic material that imparts additional microhardness to composites [31,40]. Figure 6 also represents that HA is present at the grain boundaries. The ceramic phases of HA and MgO were observed in the XRD analysis (Figure 5), which can be the primary reason for the increase in microhardness values. Nano-HA particles are present at the grain boundary of Mg. HA percentage at the grain boundary increases with its content in the matrix, which can be an obstacle in dislocation movement during the indentation of BMMCs. Subsequently, the microhardness of the composite with the addition of HA is relatively higher than that of AZ31-0HA.

Figure 9 shows the compressive strength and Young’s modulus of the BMMCs. Noticeably, compressive strength and Young’s modulus improved in BMMCs because of the presence of the HA as reinforcement. Moreover, the highest compressive strength was observed with AZ31-15HA of 198.08 ± 10.07 MPa and Young’s modulus of 39.86 ± 2.12 GPa. The lower compressive strength and Young’s modulus of AZ31-20HA compared to AZ31-15HA are due to the HA particle agglomeration in composites resulting in reduced relative density. The agglomeration of nano-HA particles was observed in SEM micrographs at higher magnifications up to 400 nm (Figure 6). The mechanical properties of AZ31-15HA were observed in the range of the human cortical bone [41].

### 4.3. Biodegradation Behavior

The degradation behavior of developed BMMCs was measured using an immersion test in the HBSS solution at 37 ± 2 °C with 7.2 ± 0.2 pH for 24 h. The degradation of the samples occurs through electrochemical reactions resulting in Mg^2+^, Ca^2+^ and PO_4_^3−^ ions leaching out into the surrounding solution. The relative weight loss was measured using Equation (1) after 24, 72 and 168 h, and is shown in Figure 10a. AZ31-0HA sample shows the highest weight loss in the HBSS solution after 24, 72 and 168 h. It happened because of the leaching out of Mg^2+^ ions from the AZ31-0HA sample surface in the anodic oxidation region. Mg^2+^ ions combined with OH^−^ (present at the cathode) and formed magnesium hydroxide (Mg(OH)_2_), which is called brucite [42]. Further, Mg(OH)_2_ reacted with Cl^−^ ions present in HBSS and formed more soluble MgCl_2_ [43]. The formation of soluble MgCl_2_ accelerated the degradation rate and resulted in the maximum weight loss in AZ31-0HA. The chemical reactions at the anode and cathode are given by Equations (4) to (7).
(4)$$\mathrm{Anodic}\;\mathrm{region}:\;\mathrm{Mg}\to {\mathrm{Mg}}^{2+}+2e^{-}$$
(5)$$\mathrm{Cathodic}\;\mathrm{region}:\;2H_{2}O+2e^{-}\to H_{2}↑+2{\mathrm{OH}}^{-}$$
(6)$${\mathrm{Mg}}^{2+}+2{\mathrm{OH}}^{-}\to \mathrm{Mg}{\left(\mathrm{OH}\right)}_{2}\left(s\right)$$
(7)$${\mathrm{Mg}}^{2+}+2{\mathrm{Cl}}^{-}\to {\mathrm{MgCl}}_{2}\left(s\right)$$

Conversely, relative weight loss was lower in BMMCs than in the pure AZ31-0HA sample. AZ31-15HA BMMC showed the lowest value of weight loss after 24 h. This means it has good integrity in HBSS and leaches out fewer ions from the sample surface during the immersion period. In addition, after 72 and 168 h, it shows weight gain that is also confirmed by the XRD and EDX mapping of the AZ31-15HA sample after 168 h of immersion test. The new phases of Mg(OH)2 and Ca(OH)2 were observed that protect against further corrosion and weight loss. Moreover, in BMMCs, the formation of the HA layer at the surface may be the reason for low weight loss. Ca^2+^ and PO_4_^3−^ ions released from the composite samples increased the degree of supersaturation of HBSS during the immersion test. The HA layer can easily grow by consuming Ca^2+^ and PO_4_^3−^ ions and OH^−^ ions in the surrounding solution. This apatite layer can form in a short time and be the reason for reducing the corrosion rate and increasing the bioactivity of biocomposites [44]. AZ31-20HA showed an almost constant weight loss during 24 and 72 h, but after 168 h, it showed weight gain. Lower densification of AZ31-20HA could be the reason for such weight loss and weight gain, which facilitated the leaching out of Mg^2+^ ions from the sample along with Ca^2+^ and PO_4_^3−^. However, after the formation of (Mg(OH)_2_) and apatite layers, the weight loss stopped and these layers could be the reason for weight gain.

Figure 10b shows the pH of HBSS after 24, 72 and 168 h of immersion test. The bulk solution of the HBSS was used for measuring pH value. When it was compared to other BMMCs, in AZ31-15HA BMMC, a lower pH value was recorded. It could be because there was a sufficient amount of HA content to retard the release of Mg^2+^ ions compared to AZ31-0HA, along with the formation of an apatite layer at the sample surface. AZ31-10HA exhibited the maximum pH value, i.e., due to diffusion of Mg^2+^, Ca^2+^ and PO_4_^3−^ ions from the sample. The formation of MgCl_2_ in the AZ31-0HA can increase the pH value after the immersion test. In the case of AZ31-20HA, the pH value could be increased due to the release of magnesium, calcium, or phosphate ions since the presence of Ca^2+^ and HPO_4_^2−^ ions also enhanced the solution’s pH value due to the addition of more bases [44].

ICPMS was used to determine how many Mg^2+^ ions were released during immersion in HBSS for 24 h. Figure 10c depicts the concentration of the released Mg^2+^ ions present in HBSS after 24, 72 and 168 h of immersion test. This test is intended to mimic their performance in an actual working environment. Results from the immersion test indicate that AZ31-15HA performs better in the in vitro test than other compositions.

It could be realized due to the formation of Mg(OH)_2_ and hydroxyapatite corrosion product layers at the sample surface. These corrosion products are deposited over the sample and behave like protective layers against further corrosion [45,46,47]. These protective layers make the degradation rate slow. Consequently, it is summarized that AZ31-15HA could be the most suitable composition of developed BMMCs that may facilitate the healing process during implantation in the human body with a growing apatite layer that helps in integration with the human bone.

Figure 11 illustrates the XRD spectra of AZ31-15HA BMMC after 168 h of immersion test in HBSS. The XRD results show the presence of new phases, such as Mg(OH)_2_ and Ca(OH)_2,_ after the immersion test. These results also confirm the formation of brucite and calcium hydroxide layers at the surface, preventing the sample from further corrosion. Both layers are insoluble and act like corrosion-protective layers. These layers can also be observed from the EDX elemental mapping, as shown in Figure 12. It is clearly seen from EDX elemental mapping that after 168 h, Mg, O, P, Ca and Zn are detected, along with Na, Cl and K. All these elements were deposited on the sample surface due to the surrounding solution that contained these elements. These results also confirm the presence of protective layers at the surface that controlled the degradation rate of AZ31-15HA and fewer Mg^2+^ ions in the bulk HBSS solution after 168 h.

Electrochemical corrosion tests were also carried out to measure the polarization resistance and electrochemical behavior of developed BMMCs. Figure 13 shows the potentiodynamic polarization curves of BMMCs recorded in HBSS at 37 ± 2 °C and 7.2 ± 0.2 pH in open-to-air conditions. These plots display two different portions: anodic polarization and cathodic polarization. The anodic polarization curve represents the dissolution of Mg^2+^ ions from the sample and the formation of the protective layers. In contrast, the cathodic polarization curve represents the rate of cathodic reaction (in this case, oxygen reduction reaction) [46]. The intersection point of anodic and cathodic Tafel lines in the potentiodynamic polarization curve is the primary point for the determination of corrosion current (I_corr_) and corrosion potential (E_corr_). Further, the corrosion rate concerning weight loss/gain analysis was calculated from Equation (3), which depicts a high corrosion rate (representing low corrosion resistance), as shown in Table 4. It was observed that the BMMCs exhibit a negative corrosion rate. The XRD, SEM, and EDS analyses revealed that the formation of passive layers of corrosion products caused a net weight gain in all BMMCs except AZ31-0HA. The potentiodynamic curve of AZ31-0HA shows the breakdown potential, which indicates the existence of localized corrosion and the curve shifted in the anodic direction, which means more Mg^2+^ ions were released from the sample. The electrochemical corrosion test confirms that the pure AZ31-0HA sample exhibits pitting corrosion at the exposed surface.

Similarly, AZ31-10HA BMMC also shows a more anodic region in the potentiodynamic curve that confirms the high Mg^2+^ ions concentration in immersing solution after 24, 72, and 168 h. In contrast, AZ31-15HA and AZ31-20HA BMMCs have more cathodic region than anodic region, indicating more cathodic reactions and hydrogen generation. These results show a good correlation with immersion test results. It was seen that these samples (AZ31-15HA and AZ31-20HA) discharged Mg^2+^ ions as well as Ca^2+^, which formed layers of Mg(OH)_2_ and Ca(OH)_2_ as corrosion products [46]. This is due to changes in the value of open circuit potential (OCP); since the test parameters were set with the position of the OCP, it consequently allowed for more dissolution of the material. Hence the samples with the more anodic OCP were not subjected to prolonged anodic polarization. The AZ31-15HA has the most noble corrosion potential value (E_corr_) amongst the samples tested, while the AZ31-15HA and AZ31-20HA samples show high corrosion current densities. The reason for showing high corrosion current density could be the higher percentage of the HA on the surface available for corrosion. Figure 6 shows the distribution of HA nanopowder particles surrounding the Mg grains at the surface. Table 4 represents the E_corr_, I_corr_ and corrosion rate of BMMCs (from weight loss/gain analysis), which indicate that AZ31-15HA is the optimum composition among the compositions considered.

## 5. Conclusions

In this study, AZ31-HA BMMCs were fabricated through microwave sintering using a multimode microwave applicator at 2.45 GHz and 650 W maximum power. The mechanism of microwave sintering and time-temperature profile with targeted materials has shown that microwave material processing is a material-dependent technique and interacts differently with different materials. Moreover, the sintered BMMCs were characterized for their phase composition, sintered density, microhardness, compressive strength, Young’s modulus, and corrosion behavior in an in vitro environment. The major conclusions are:AZ31-0HA showed the highest relative density. It decreased with increased HA content. The addition of hard HA particles hindered the densification of the BMMCs as it was difficult to press during compaction and with a large difference in the sintering temperature of AZ31 metal alloy and HA.The microhardness of the BMMCs increased with increasing HA content owing to the presence of HA hard phase in the composite. It was uniformly distributed in the composite volume that was observed in microstructures.The highest compressive strength and Young’s modulus were recorded with the AZ31-15HA. Further HA addition decreased the compressive strength and Young’s modulus. It is attributed to a decrease in the density and agglomeration of HA particles.The AZ31-15HA exhibited the least weight loss for 24, 72 and 168 h immersion tests, and lower pH change of HBSS and discharge of Mg^2+^ ions after immersion tests in HBSS.The corrosion resistance of AZ31-15HA is the best among the four compositions. It exhibited weight gain after 72 h, resulting in a negative (−3.25 mm/year) corrosion rate. A negative corrosion rate indicates passive corrosion because of the formation of protective layers of Mg(OH)_2_ and Ca(OH)_2_ layers at the surface. It was also observed in electrochemical impedance spectroscopy (EIS) results and these results will be reported in a future study.The developed biodegradable composites in this study can be considered for human body fixation aids as they show favorable properties for orthopedic application. The BMMCs in this study did not show any toxicity during the in vitro cytotoxicity test.

## Figures and Tables

**Figure 1 materials-16-01905-f001:**
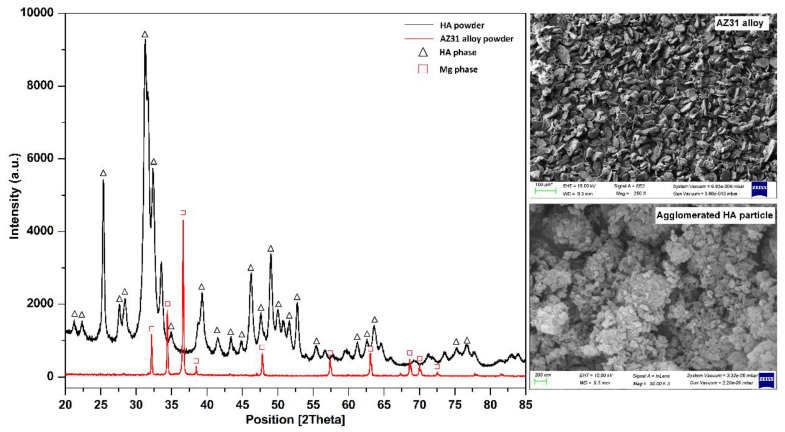
XRD spectra of AZ31 alloy and HA powder (Inset: SEM micrograph of AZ31 alloy powder particles (100 µm scale) and agglomerated HA powder particles (200 nm scale)).

**Figure 2 materials-16-01905-f002:**
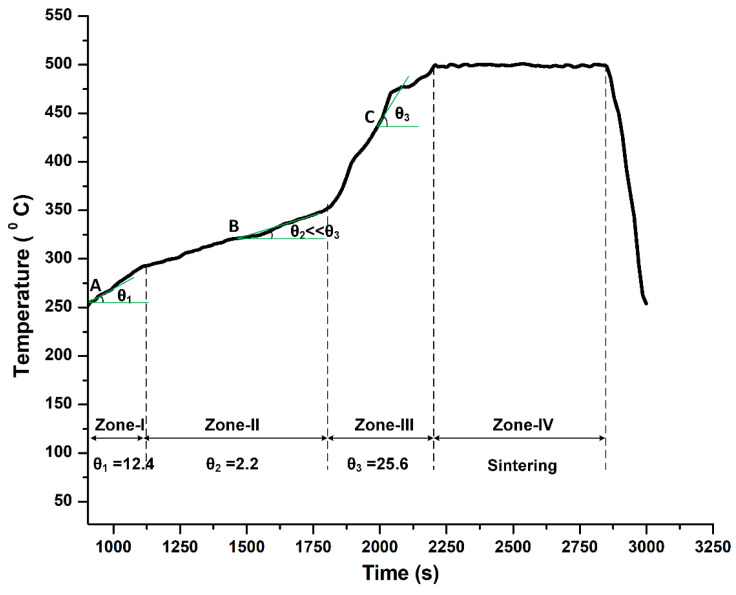
Time–temperature curve in microwave sintering of AZ31-15HA BMMC.

**Figure 3 materials-16-01905-f003:**
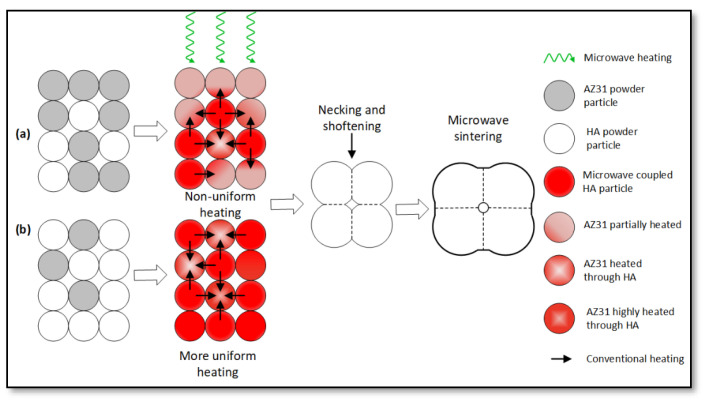
Microwave sintering mechanism and the interaction of microwave radiation with (**a**) AZ31 metal alloy and small amount of HA powder particles and (**b**) AZ31 metal alloy with large amount of HA powder particles.

**Figure 4 materials-16-01905-f004:**
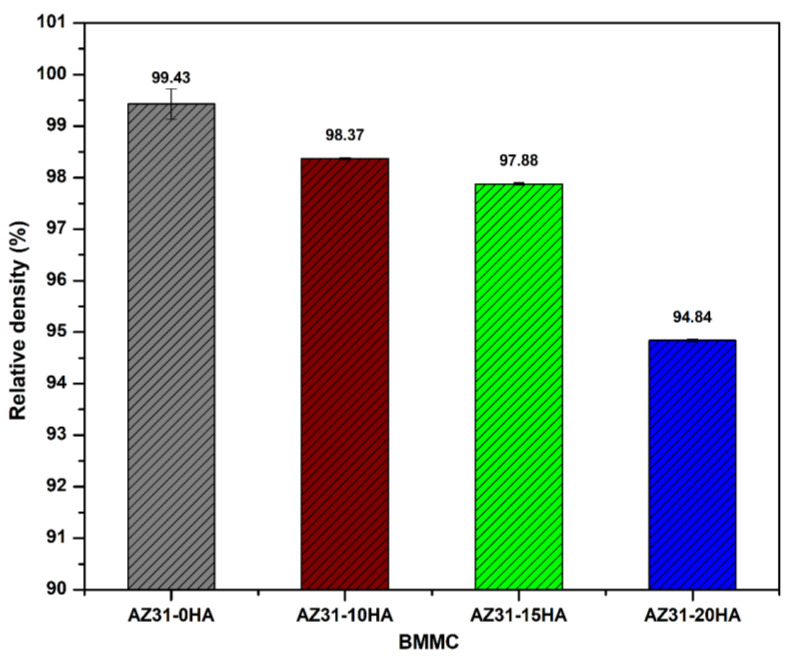
Relative density of microwave-sintered AZ31-HA BMMCs.

**Figure 5 materials-16-01905-f005:**
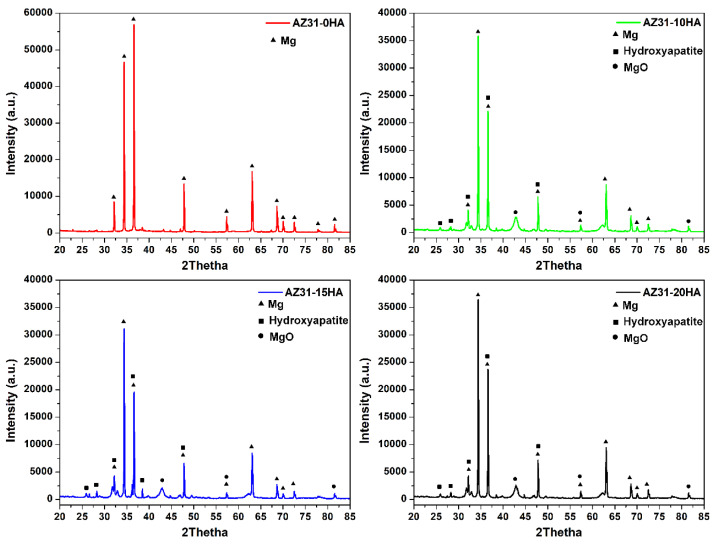
Typical XRD spectra of microwave-sintered AZ31-HA BMMCs.

**Figure 6 materials-16-01905-f006:**
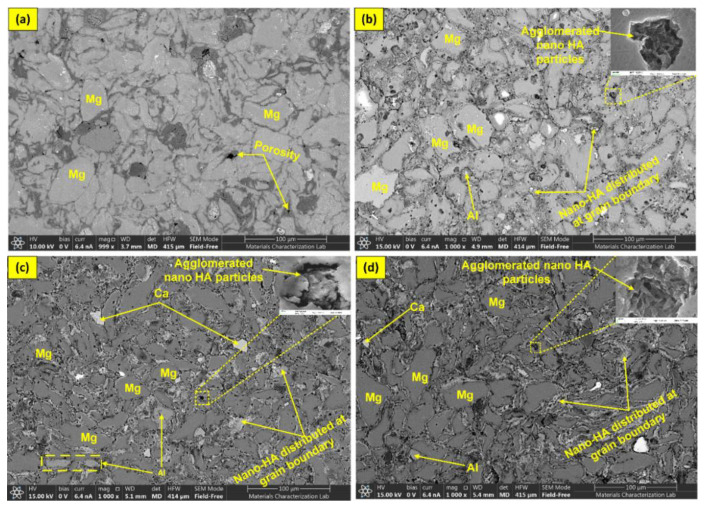
Backscattered electron microscopic SEM image of sintered BMMCs; (**a**) AZ31-0HA, (**b**) AZ31-10HA, (**c**) AZ31-15HA and (**d**) AZ31-20HA.

**Figure 7 materials-16-01905-f007:**
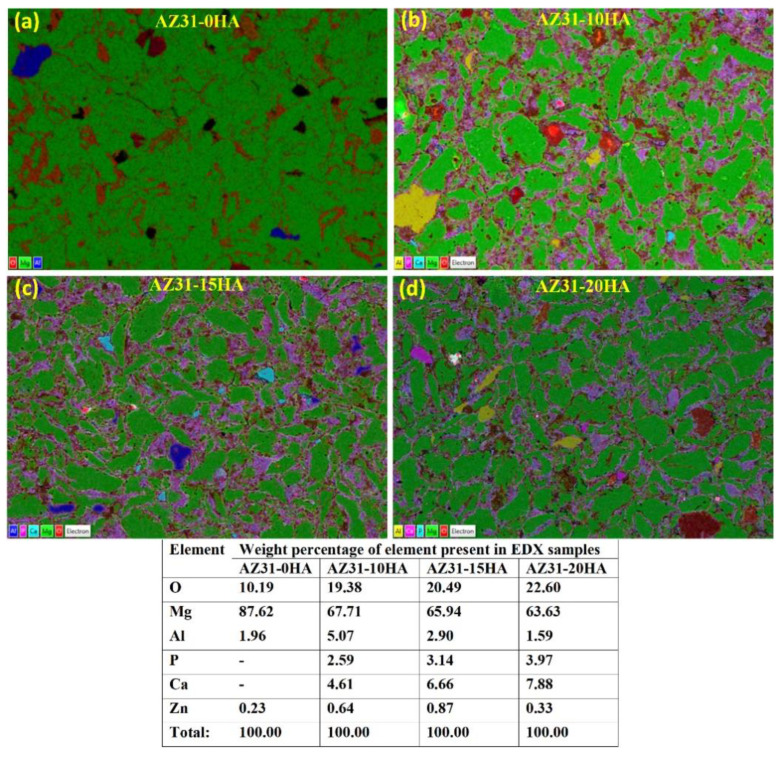
Elemental mapping of AZ31-HA BMMCs from energy-dispersive X-ray spectroscopy along with percentage of element present in the EDX samples (scale: 100 µm).

**Figure 8 materials-16-01905-f008:**
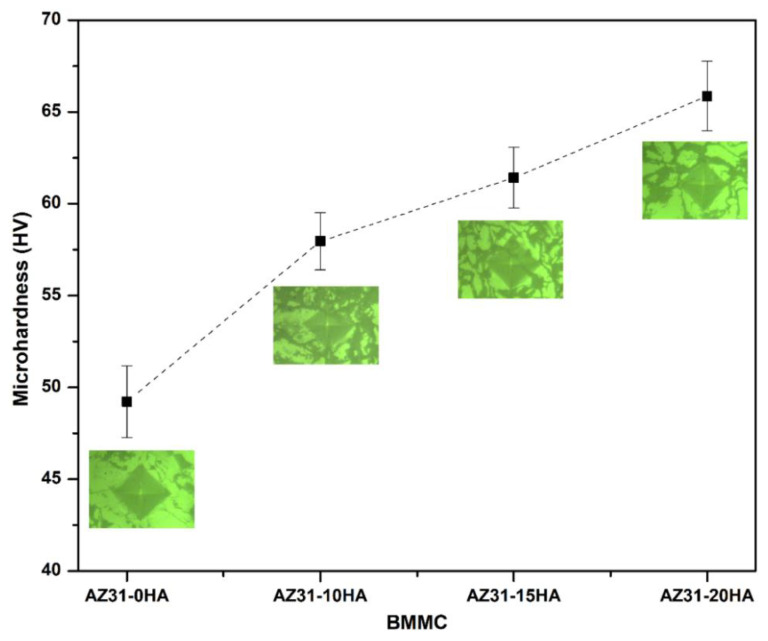
Typical microhardness variation of microwave-sintered AZ31-HA BMMCs; (Inset: optical image of indentation (scale: 200 µm)).

**Figure 9 materials-16-01905-f009:**
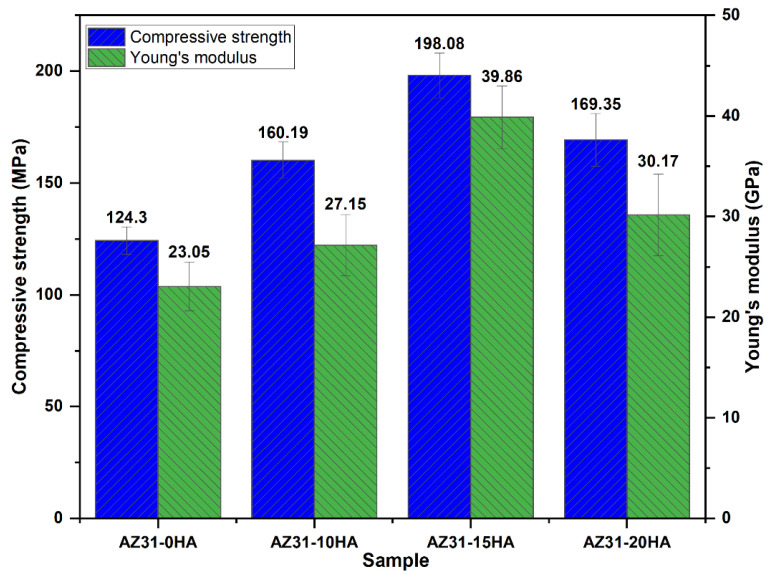
Compressive strength and Young’s modulus of microwave-sintered BMMCs.

**Figure 10 materials-16-01905-f010:**
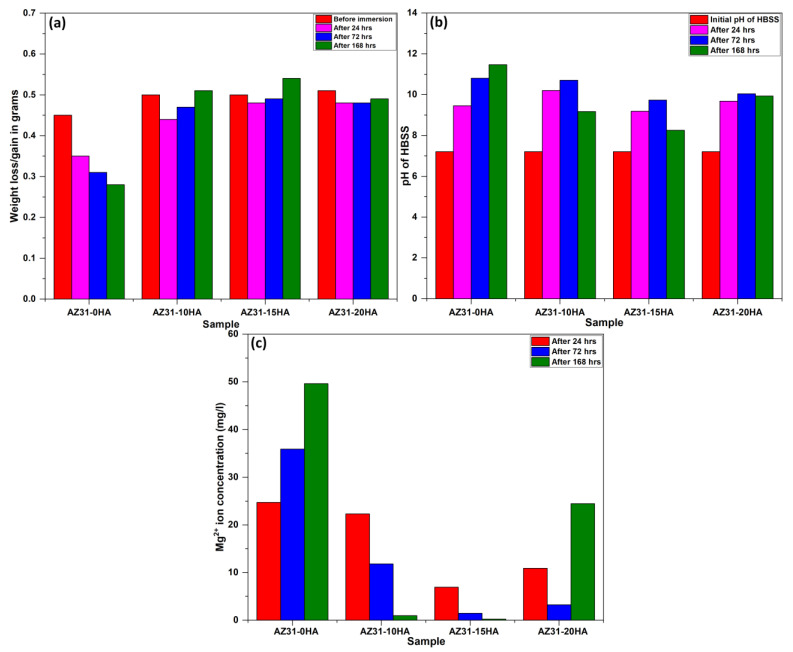
Results of immersion test in HBSS after 24, 72 and 168 h of exposure time (**a**) relative weight loss, (**b**) changes of pH value of HBSS and (**c**) changes of Mg^2+^ ion concentration in HBSS.

**Figure 11 materials-16-01905-f011:**
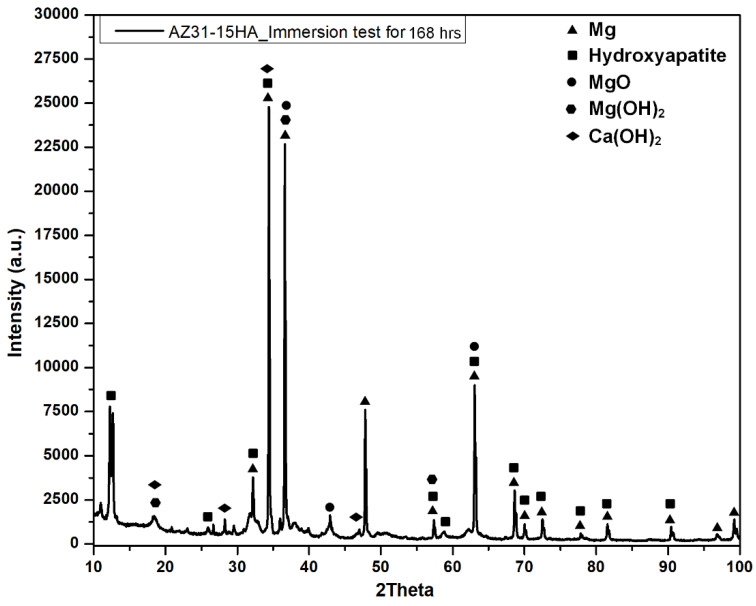
XRD spectra of AZ31-15HA BMMC after immersion test for 168 h.

**Figure 12 materials-16-01905-f012:**
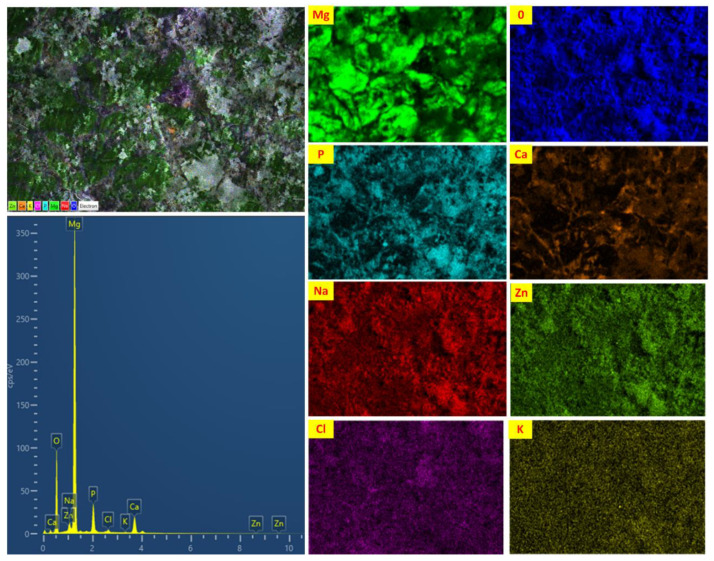
EDX analyses; electron mapping of elements present at AZ31-15HA BMMC surface after immersion test in HBSS at 37 ± 2 °C for 168 h.

**Figure 13 materials-16-01905-f013:**
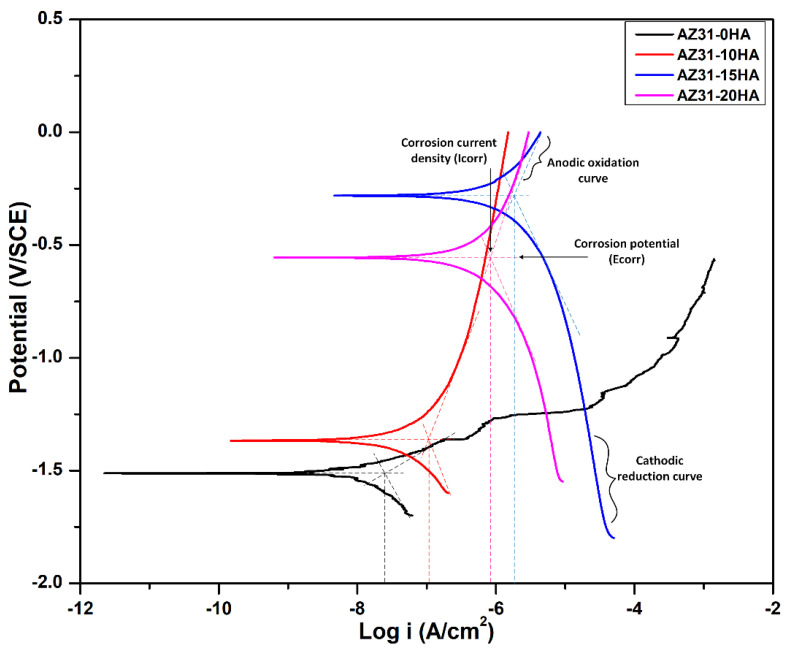
Electrochemical potentiodynamic curve of the BMMCs in HBSS at 37 ± 2 °C with SCE as a reference electrode.

**Table 1 materials-16-01905-t001:** Chemical composition of AZ31 alloy.

Main Elements	Magnesium	Aluminum	Zinc	Manganese	Silicon
Weight percent (wt.%)	95–97	2.5–3.5	0.6–1.4	0.2	0.1

**Table 2 materials-16-01905-t002:** Details of processing conditions used in microwave sintering of AZ31-HA BMMCs.

Parameter	Value
Frequency	2.45 GHz
Microwave power	330–650 W
Crucible material	Alumina
Insulation material	Alumina wool
Sample compositions	AZ31-0HA, AZ31-10HA, AZ31-15HA and AZ31-20HA
Sintering holding time	10 min
Sintering temperature	500 °C
Processing condition	Forming gas environment (95% nitrogen + 5% hydrogen)

**Table 3 materials-16-01905-t003:** Ionic concentration of chemicals used in HBSS preparation [37].

Solution	Ionic Concentration (mmol/L)							
	Na^+^	K^+^	Ca^2+^	Mg^2+^	Cl^−^	HCO_3_^−^	HPO_4_^2−^	SO_4_^2−^
**Blood plasma**	142.0	5.0	2.5	1.5	103.0	27	1.0	0.5
**HBSS**	141.6	5.81	1.26	0.81	144.8	4.09	0.78	0.81

**Table 4 materials-16-01905-t004:** Corrosion potential, corrosion current density and corrosion rate of BMMCs in HBSS.

Samples	E_corr_ (V)	I_corr_ (A/cm^2^)	Corrosion Rate after 168 h Immersion Test (mm/Year)
AZ31-0HA	−1.51	2.9 × 10^−8^	8.48
AZ31-10HA	−1.38	1.3 × 10^−7^	(−)0.89
AZ31-15HA	−0.28	1.4 × 10^−6^	(−)3.25
AZ31-20HA	−0.55	1.6 × 10^−6^	0.82

## Data Availability

Not applicable.

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
