# Peer review of "Characterization of AZ31/HA Biodegradable Metal Matrix Composites Manufactured by Rapid Microwave Sintering"

_materials, 2023, doi:10.3390/ma16051905_

Round 1

Reviewer 1 Report

Dear editors and writers:

The paper “Characterization of AZ31/HA biodegradable metal matrix composites manufactured by rapid microwave sintering” has been carefully reviewed. The research has great reference significance. However, there are several points that should be addressed before this article is considered for publication:

1. XRD results are not calibrated. The peak corresponds to phase and crystal plane are not given. This analysis needs to be supplemented.

2. The picture in Figure 8 is wrong. The data of two different units cannot share the same coordinate axis, and should be modified to the form of double coordinate axes.

3. In general, the elastic modulus is proportional to the volume fraction of the reinforcement. The increase in elastic modulus in Figure 8 does not seem to be common sense. When the HA content increases from 0 to 10, the elastic modulus increases by 4.1 GPa. When the HA content increases from 10 to 15, the elastic modulus increases by nearly 12 GPa. The decrease of material density in the two stages is close. What causes the change of elastic modulus?

4. The abscissa of Figure 7 is misleading, and the lengths of 0-10 and 10-15 are exactly the same, which cannot reflect the true law of the data. Should be changed to a linear abscissa.

5. Equation 7 is not balanced.

6. The annotation in Figure 10 is written 168h, while the picture is marked 24h.

7. The clarity of Figure 11 is too low to see the content. And it 's best to give an organizational comparison before and after immersion.

8. The ions should be written as “Mg2+”, and the labeling method in this paper seems not very common.

In summary, there are some problems in the paper, which need to be modified and supplemented. It is recommended that it be published after revision.

Reviewer 2 Report

The work discussed the subject of the Characterization of AZ31/HA biodegradable metal matrix composites manufactured by rapid microwave sintering Presented in the manuscript research shows a decent scientific, and by above and taking care of the high level of research target to Materials, I recommend only a major revision of this article.

General

The conclusion of the article points out its t weaknesses which for the obtained results question its significance. In the reviewer's opinion, the main goal of the work could be reached if the authors decide to develop the processing range. The initiative to sinter the MMC by MW infrastructure is not new, however, the range of interesting phenomena during processing still could be explored and put a new value into the material subject. For example, fig 2 represents the generally used curve for all the systems which were sinter or typical ones. We do not know. Why the authors used this kind, of course, nothing here is justified. Just we made it like that.

Obtained differences between the samples remain obvious and predictable maybe a potentiodynamic test stays interesting however the authors do not focus on the potential range movement and a negative corrosion rate (from tab 4) which stays also omitted.

Reviewer 3 Report

1. It is suggested to label the phase species on the XRD curve peaks in Figure 1.

2. The AZ 31 / AH composites prepared by rapid microwave sintering have better mechanical properties and biodegradability, a comparison of the approved similar composite material should be selected in the main text, in order to further illustrate the composite method used and gain performance advantages or need further improvement in this paper.

 The main question of this paper is the preparation of AZ 31 / HA magnesium alloy by rapid microwave sintering. The original is reflected in the combination of rapid microwave sintering method and AZ 31 / HA material. The performance of this material is not compared to the other methods, which needs to be added. It cannot be shown that this research work addresses the specific gaps in such materials, but there is some progress in some properties, such as corrosion. However, the preparation method should be improved in the material microstructure uniformity, mainly to further control the size uniformity of Mg phase. The conclusion of the article is basically consistent with the presented evidence, addressing the question of this material organization and performance characterization. The references are basically appropriate, and the individual figures are not complete, as shown in Figure 1.

Round 2

Reviewer 1 Report

Accept in present form

Author Response

Thank you so much for accepting our manuscript.

Reviewer 2 Report

The work discussed the subject of the Characterization of AZ31/HA biodegradable metal matrix composites manufactured by rapid microwave sintering Presented in the manuscript research needs a major revision to improve the quality of the manuscript.

General

The conclusion of the article points out its weaknesses, which for the obtained results question its significance. In the reviewer's opinion still, the main goal of the work could be reached if the authors decide to develop the processing range analysis. The initiative to sinter the MMC by MW infrastructure is not new, however, the range of interesting phenomena during processing still could be explored and put a new value into the material subject. For example, fig 2 represents some curve of some sample the reader don’t know which and cannot compare them to see the difference. The reader don’t know why the authors use this track for processing and for example also needs to guess why the authors increase the microwave power during sintering and when it was done?

Obtained differences between the samples remain obvious and predictable maybe a potentiodynamic test stays interesting, however the authors do not focus on the potential range movement and a negative corrosion rate (from tab 4) which still  need an explanation and additional research after test.
